# Optimal Irrigation Scheduling for Greenhouse Tomato Crop (*Solanum Lycopersicum* L.) in Ecuador

Javier Ezcequiel Colimba-Limaico * , Sergio Zubelzu-Minguez and Leonor Rodríguez-Sinobas

Grupo de Investigación Hidráulica del Riego, Universidad Politécnica de Madrid, Av. Puerta de Hierro, no 2–4, Ciudad Universitaria, 28040 Madrid, Spain; sergio.zubelzu@upm.es (S.Z.-M.); leonor.rodriguez.sinobas@upm.es (L.R.-S.)
* Correspondence: javier.colimba.limaico@alumnos.upm.es; Tel.: +593-9-99571363

**Abstract:** Tomato crop is grown worldwide and is considered a mass consumer product. In Ecuador, tomato growers face two major issues: water scarcity and water mismanagement, which cause a reduction in the framers' gross income and ecosystem services. This paper is aimed at finding an optimal irrigation scheduling in greenhouse tomato crop to achieve a balance among production, fruit quality and water use efficiency. Thus, two experiments were settled. In the first experiment, four water doses (80, 100, 120 and 140% ETc) and two irrigation frequencies (one and two irrigations per day) were compared. The second experiment evaluated the two best water doses of the first one (100 and 120% ETc) and four irrigation frequencies (one and two irrigations per day, one irrigation every two days, one irrigation every three days). Each experiment monitored the variables for tomato production (plant height, stem diameter, fruits per plant, yield) and tomato quality (pH, total soluble solids, titratable acidity). The study concluded that water doses affected more than irrigation frequency to fruit quality and production. The dose of 100% ETc, applied in one irrigation per day, is suggested to obtain a balance between production, fruit quality and water use efficiency.

**Keywords:** water dose; irrigation frequency; drip irrigation; tomato quality; tomato production; water use efficiency

## 1. Introduction

Tomato (*Solanum lycopersicum* L.) is one of the major horticultural crops consumed [1,2] and cultivated [3] worldwide. Its largest harvested area concentrates in China, Nigeria and India [4]. In Ecuador, the harvested tomato area was 2579 ha in 2020 with an average yield about 14.9 t ha$^{-1}$ and a total production about 38,438 t [4]. The largest production area is located in the Sierra Region (1976 ha) and the provinces with the largest production in 2020 were Imbabura, Manabí and Pichincha with 971 ha, 579 ha and 263 ha, respectively [5]. In Ecuador, the major problems in tomato crop are mismanagement of water and fertigation [6], virus diseases [7,8] and plagues such as Tuta absoluta [9]. In addition, low soil fertility and/or contamination, as well as water quality and drought are other issues that not only affect tomato but agriculture production, in general [10].

Tomato crop water requirements are high [11–14]. Water is a key element for crop yield under water deficit conditions [15] but it is a scarce resource which limits crop productivity and quality [16]. Hence, water shortage affects farmer's income, especially in arid and semi-arid areas [17]. Likewise, it is foreseen that about 50% of the world's population will live in water scarcity regions in 2050 [18].

In Ecuador, irrigated agriculture demands 80% of the total available water [19] and it is estimated it will increase 22.4%, between 2010 and 2025 [20]. At present, water resources are limited and conflicts among water users are expected in the short term [20]. In addition to water scarcity, farmers mismanage the resource.

Farmers always blindly overwater greenhouse tomato to achieve high yield [21]. However, an increase in water supply does not increase yield proportionally [22]. Likewise,

tomato producers in the area over-irrigate their crops to increase yield which not only leads to wastewater but also worsens the balance between supply and water demand [23].

The application of water to the crops, either in excess or in deficiency, can cause serious alterations in the plant. Thus, water deficit causes water and nutritional stress and reduces biomass production and marketable yield [24]. On the contrary, the excessive application of water, in addition to causing waste, also promotes nitrogen leaching, the emission of nitrous oxide and soil salinization [25], on the other hand, applying too much water in irrigation it tends to cause an excessive accumulation of biomass in tomato plants, which reduces the yield [3,26]. Therefore, it is imperative to optimize the amount of water and apply it at an appropriate irrigation frequency to achieve a balance between plant growth and yield to improve water use efficiency WUE.

The adoption of strategies to save water and maintain or improve WUE have become a priority [27]. Many reports cope water scarcity with water deficit irrigation [28] and other studies focus on increasing WUE with the application of water doses lower than tomato water requirements. However, deficit irrigation enhances soil moisture deficit in several horticultural crops such as tomato, and has resulted in yield reduction [29], showing that high WUE and crop production will never be simultaneous [3], which is detrimental to the farmer's economy.

Irrigation mismanagement affects water storage in reservoirs [30] and can be ameliorated if water dose and irrigation frequency are properly determined [31,32]. Irrigation frequency affects, among others, soil moisture distribution, nutrient mobility, soil salinity, crop yield and WUE. Ref. [33] argue that proper irrigation frequency can balance soil moisture and oxygen concentration within the root zone throughout the growing season. A high irrigation frequency allows to obtain better plant development [34], higher yield [35–37], greater amount of total soluble solids (TSS) [26,37] and better WUE [26]. On the other hand, very high irrigation frequencies keep soil surface close to saturation thus, evaporation losses are higher [38], likewise can induce soil salinity and/or hypoxia if infiltration is low [39]. Contrary [40] obtained the highest yield and WUE in the largest intervals (7 and 9 days) and argued that these frequencies improved root development. As a summary, research community does not agree on the effect of irrigation interval in WUE [33].

Several reports have focused on the effect of water doses and irrigation frequency in tomato crop although, in most of them, both variables were studied separately. Therefore, in order to guarantee a given crop production (quantity and quality), farmer's income and irrigation efficiency, the study of irrigation strategies for a proper application of water under a practical irrigation frequency would be desirable. This study is aimed at finding an optimal irrigation schedule for tomato crop, which balances production, fruit quality and WUE. It is foreseen that the results will help local famers to make sustainable decisions on their irrigation practice.

## 2. Materials and Methods

### 2.1. Site Description

The study was carried out in a greenhouse in the town of Natabuela ($0°20'16.67''$ N, $78°12'0.65''$ W; 2430 m.a.s.l.), Imbabura Province (Ecuador) from September 2019 to December 2020. The area has a temperate climate, the annual average temperature and precipitation are 15 °C and 635 mm, respectively. The physical and chemical properties of soil were measured within the 0–20 cm layer before plant transplanting (August 2019). The soil was sandy loam texture with the following values: organic matter = 3.2%, bulk density = 1.22 $g\,cm^{-3}$, pH = 7.77, field capacity = 34.81% and permanent wilting point = 14.31%.

### 2.2. Crop Management

The greenhouse covers 355.25 $m^2$ (24.5 m × 14.5 m), one experimental plot occupies 8.96 $m^2$ (6.4 m × 1.4 m). A tomato's row was placed on each plot and contained 16 tomato plants with planting density of 17,857 plants $ha^{-1}$ (1.4 m × 0.4 m). The tomato hybrid Pietro

(HM. CLAUSE, Davis, CA, USA) was selected since it is the most cultivated in Imbabura Province. The study performed two experiments and tomato plants were transplanted on 6 September 2019 and on 12 June 2020 in the first and second experiment, respectively.

Fertilization was performed according to the soil analysis. In the first experiment, tomato plants were fertilized twice (30 and 70 days after transplanting) and 175 kg ha$^{-1}$ total nitrogen, 65 kg ha$^{-1}$ phosphorus ($P_2O_5$) and 234 kg ha$^{-1}$ potassium ($K_2O$) were applied [41]. In the second experiment, four fertilizers' applications were supplied: the first before transplanting and the rest 40, 80 and 110 days after transplanting. The number of fertilizers was: 400 kg ha$^{-1}$ total nitrogen; 200 kg ha$^{-1}$ phosphorus ($P_2O_5$) and 600 kg$^{-1}$ potassium ($K_2O$) [42].

In the first experiment, fertilizer doses were lower since they were applied over three years to fallow soil, which is high in nutrient concentration due to the frequent application of organic matter. In the second experiment, fertilizer doses increased to supply the nutrient plant consumption.

### 2.3. Experimental Design

The present study focusses on finding an optimal irrigation schedule for tomato crop that balances crop production and quality. Thus, several water doses and irrigation frequencies were assessed. The first experiment monitored two irrigation frequencies: two irrigations per day (F1) and one irrigation per day (F2), as well as four water doses: 80% $ET_c$ (L1), 100% $ET_c$ (L2), 120% $ET_c$ (L3) and 140% $ET_c$ (L4-control-local farmer dose). The second experiment evaluated four irrigation frequencies: two irrigations a day (F1), one irrigation a day (F2), one irrigation every two days (F3) and one irrigation every three days (F4), as well as two water doses (the best of the first experiment): 100% $ET_c$ (L1) and 120% $ET_c$ (L2).

The first experiment conducted a factorial arrangement (2 × 4) in split plots under a completely randomized block design with four replicates. The second experiment conducted a factorial arrangement (2 × 4) under a completely randomized block design with four replicates.

### 2.4. Crop Water Requirement Determination

Actual evapotranspiration ($ET_c$) was calculated according to [43].

$$ET_c = ET_o \cdot K_c \tag{1}$$

where $ET_o$ is the reference crop evapotranspiration and $K_c$ is the crop coefficient. $K_c$ was determined according to [43,44] as they are shown in Table 1.

**Table 1.** Kc values used in both experiments.

| Phenological Stages | Kc First Experiment | Kc Second Experiment | Phase Duration (Days) |
|---|---|---|---|
| Initial | 0.55 | 0.55 | 35 |
| Development | 1.05 | 1.05 | 45 |
| Production | 1.15 | 1.15 | 70 |
| Final | 0.90 | 0.75 | 30 |

In the final stage of the second experiment, $K_c = 0.75$ since plants showed a lower development than in the first experiment and soil water content was higher.

$ET_o$ was determined by the evaporation pan method [45].

$$ET_o = E_{pan} \cdot K_p \tag{2}$$

where $E_{pan}$ is pan evaporation and $K_p$ is pan coefficient.

Evaporation was measured daily at 7:00 a.m. in a plastic evaporimetric pan [45] placed in the center of the greenhouse. The $K_p$ value used was 1.0 according to [46].

### 2.5. Irrigation System

The experimental design comprised of 32 cropping plots (8.96 m$^2$) irrigated with a branched irrigation network. A polyethylene manifold (32 mm) was deployed over the soil where the laterals (nominal diameter of 16 mm) were inserted. Table 2 presents the hydraulic properties of emitters which were spaced 0.20 m. Inlet pressure was supplied by a pump (0.746 kW). Likewise, each crop line was fed by two laterals and each plant received water from four emitters.

**Table 2.** Emitter's characteristics.

| Model | Nominal Flow Rate (L/h) at 0.1 MPa | $Q = KH^x$ (H in MPa) | | Manufacturer Coefficient of Variation (%) |
| | | K | x | |
| --- | --- | --- | --- | --- |
| DP Line 35MIL | 2.10 | 21.196 | 0.4754 | 2.52 |

At the upstream end of the irrigation system, a flow meter (HIDROMETERS, LXSG-40E5/RLN1, Cotia, Brazil), a pressure head gauge and a pressure regulator valve were installed.

At the beginning of each experiment, the irrigation system was evaluated at an inlet pressure of 0.1 MPa. The Karmeli and Keller uniformity coefficient [47] was 93.2% and 93.5%, respectively and the discharge variation coefficient was 3.4% and 3.0%, respectively. Thus, water application uniformity qualified as very good.

### 2.6. Measurements

The study monitored weather data inside greenhouse, soil water content, agronomic variables, fruit quality parameters and water use efficiency. The agronomic variables and fruit quality parameters were measured across the sampling area, which was representative of the plot; thus, 10 and 12 plants (located at the plot's center) were selected in the first and second experiment, respectively.

#### 2.6.1. Greenhouse Weather Data

Digital Hygro-Thermometer (Boeco, BOE 327, Hamburg, Germany) were used to measure air temperature and relative humidity (every hour) from six in the morning to six in the afternoon.

#### 2.6.2. Soil Moisture Content

Soil moisture content was measured with a portable Digital Soil Moisture Meter, YIERI, Shenzhen, China (PMS710 measurement range 0–50%, $\pm(0.5\% \, n + 2)$).

In the first experiment, soil moisture was monitored to observe its daily evolution. Five measurements were gathered along 24 h: the first before the first irrigation; the second one hour after the first irrigation; the third before the second irrigation; the fourth one hour after the second irrigation and the fifth the next day before the first irrigation. The measurements were taken at 15 cm depth and performed in two sampling zones (at the first and third section of the sampling area). Each sampling zone contains three sampling points (two laterals and one central). The measurements were made 7 cm apart from the dripper in the first, and 17 cm apart from the dripper and 15 cm from stem base in the second. The soil moisture content for each treatment was determined as the average of the six measurements.

In the second experiment, soil moisture was monitored to observe its temporal evolution. Thus, the measurements were carried out every day before watering in two points located at the fourth and at tenth plants in the sampling plot. The sampling points located

at 7 cm from the dripper and at 15 cm depth. The soil moisture content for each treatment was determined as the average of two measurements.

### 2.6.3. Agronomic Variables

The agronomic variables selected were plant height, stem diameter, number of fruits per plant and yield.

These variables were measured in all plants of the sampling plot. A measuring tape was used for plant height and the measurements were taken from the base of the stem to the insertion point of the last truss. A digital caliper measured the stem diameter at the height of the eighth truss. The number of fruits per plant was determined as the average of fruits considering all plants from the sampling plot.

After harvest, tomato fruits were weighed. The total and marketable yields were calculated at the end of each experiment. Regarding fruit weight and considering the parameters established by the Ibarra Wholesale Market (Imbabura province) fruits were divided into four categories, i.e., <70 g (very small-sized fruit), 70–100 g (small-sized fruit), 100–150 g (medium-sized fruit) and >150 g (big-sized fruit). Very small, misshapen and cracked fruits were considered unmarketable.

### 2.6.4. Quality Parameters

The quality parameters were monitored in two and three samples per experimental plot, and each sample was analyzed separately. In the first experiment, samples were taken at 119 and 130 days after planting, while in the second experiment they were taken at 133, 147 and 167 days after planting. For each sample, three fruits (with similar-size, maturity and no external defects) were picked. Sample's juice was obtained using an extractor (Homissi, PC-700, Zhejiang, China). Likewise, TSS (° Brix) was measured by a digital refractometer (HANNA, HI96801, 0–85%, Smithfield, RI, USA) and pH was measured by a pH tester (HANNA, HI98103, Nușfalău, Romania). Finally, titratable acidity (g 100 $g^{-1}$ FW, as citric acid) was measured by titration using 0.1 N NaOH against 4:1 dilution of tomato extract with water [48]. All samples were performed twice.

### 2.6.5. Water Use Efficiency

WUE was estimated as total WUE (kg m$^{-3}$) and marketable WUE (kg m$^{-3}$) as reported by [49]. The first was calculated as the ratio of total yield (kg plant$^{-1}$) and total water applied to the plant (m$^3$ plant$^{-1}$). The second was determined as the ratio of marketable yield (kg plant$^{-1}$) and total water applied to the plant (m$^3$ plant$^{-1}$).

### 2.6.6. Statistical Analysis

All measurements were analyzed by analysis of variance, using INFOSTAT statistical software, Student Version 2018 [50]. For means' comparison, Duncan's multiple range test ($p < 0.05$ significance level) was used for both factors (water dose and irrigation frequency).

## 3. Results

### 3.1. Greenhouse Weather Data

Figure 1 shows the data monitored in the greenhouse. In the first experiment, the average air temperature ranged from 17.9 to 26.5 °C (average value 21.6 °C) and relative humidity varied from 38.7 to 74.1% (average value 55.0%). In the second experiment, temperature ranged from 18.4 to 27.5 °C (average value 23 °C) and relative humidity varied from 34.0 to 69.7% (average value 50.0%).

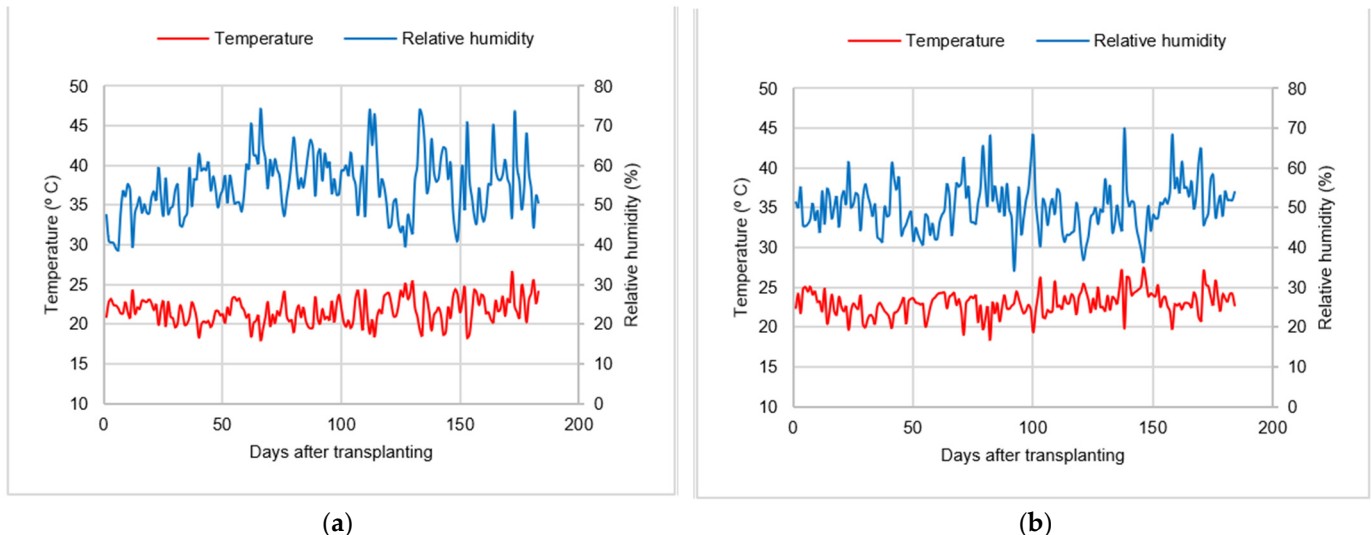

(**a**)　　　　　　　　　　　　　　　　　　　(**b**)

**Figure 1.** Average air temperature and average relative humidity values monitored inside the greenhouse. (**a**) first experiment, (**b**) second experiment.

### 3.2. Crop Water Requirements

The evaporation ($E_{pan}$) and evapotranspiration ($ET_c$) were very variable (see Figure 2) since the climatic conditions in the study area varied drastically from day to day. In the first experiment, $E_{pan}$ ranged from 0.4 to 5.5 mm day$^{-1}$ (average value 2.5 mm day$^{-1}$) and $ET_c$ varied from 0.5 to 4.7 mm$^{-1}$, (average value 2.2 mm day$^{-1}$). In the second experiment, $E_{pan}$ oscillated from 0.8 to 4.5 mm day$^{-1}$ (average value 2.7 mm day$^{-1}$) and $ET_c$ ranged from 0.7 to 4.9 mm$^{-1}$(average value 2.5 mm$^{-1}$).

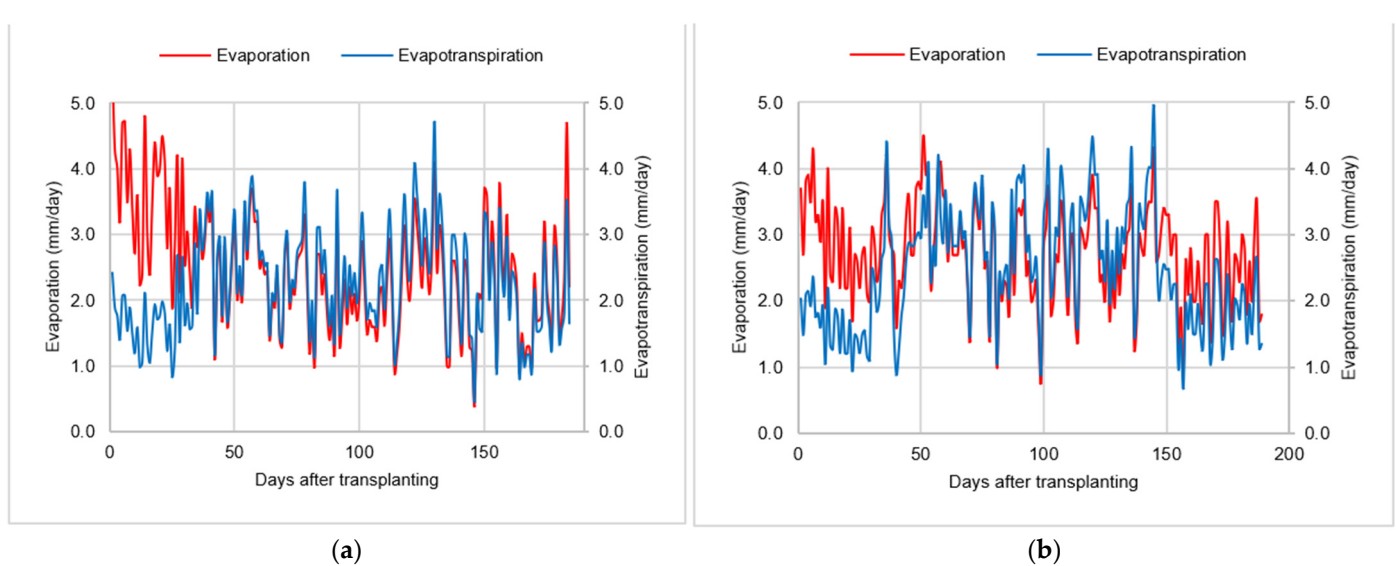

(**a**)　　　　　　　　　　　　　　　　　　　(**b**)

**Figure 2.** Daily evaporation and evapotranspiration inside the greenhouse. (**a**) first experiment, (**b**) second experiment.

### 3.3. Soil Moisture

The daily evolution of soil water content is presented in Figure 3. All the four doses showed the same trend but have differences in moisture content. Soil moisture increases abruptly just after the first irrigation, it remains stable for approximately two hours and then, it decreases gradually. For two irrigations a day, the soil moisture increases after the first irrigation and remains stable (approximately one hour) then, it decreases until the second irrigation where it increases again above the one observed at the end of the first irrigation.

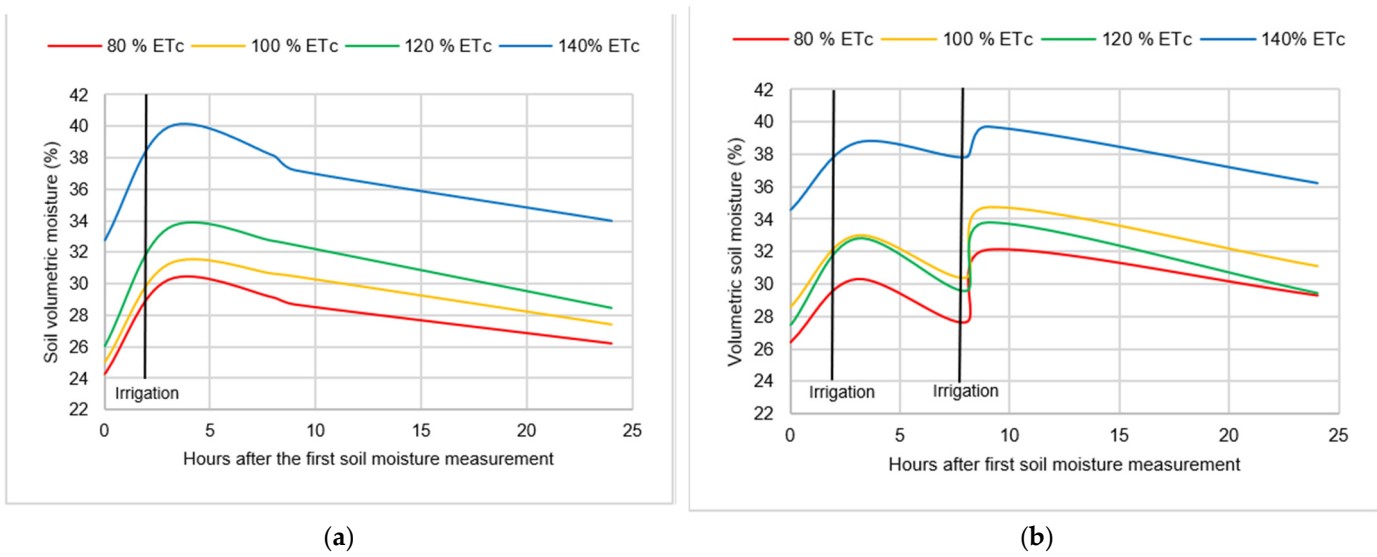

(a)                                                                                       (b)

**Figure 3.** Daily evolution of soil water content. (**a**) One irrigation a day; (**b**) two irrigations a day.

The temporal evolution of soil water content is presented in Figure 4. The trend observed in this period of time was similar for the entire crop. In both experiments, soil moisture variation was lesser at higher irrigation frequencies. On the contrary, soil water content was highly variable at lower irrigation frequencies.

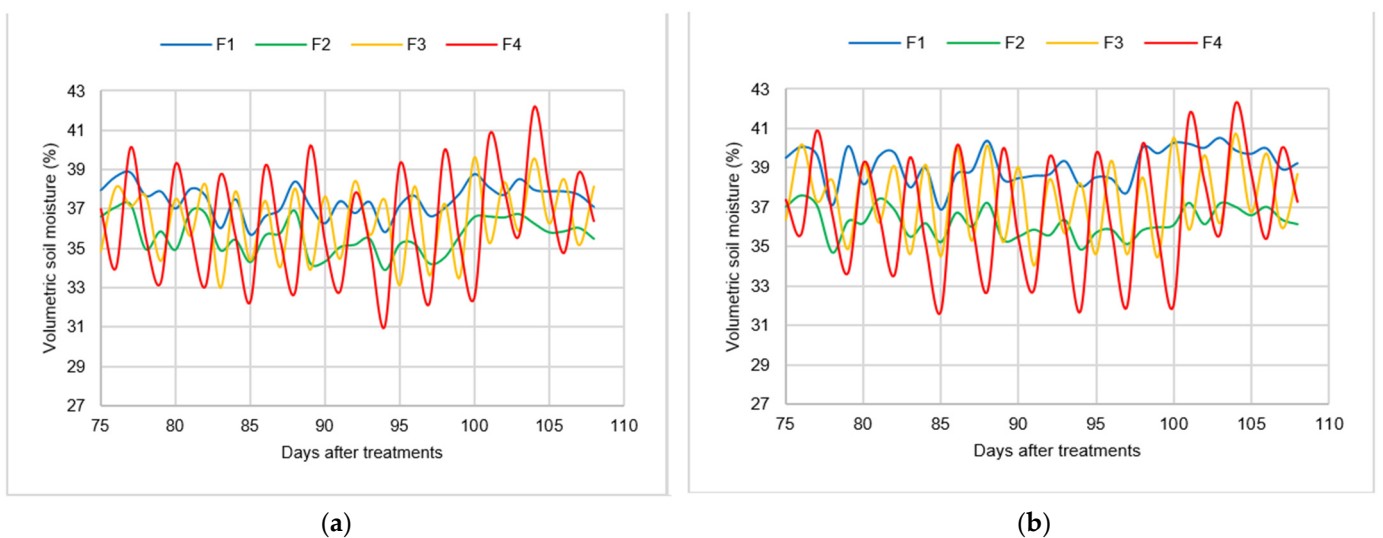

(a)                                                                                       (b)

**Figure 4.** Temporal evolution of soil moisture. (**a**) 100% ET$_c$; (**b**) 120% ET$_c$. (F1: two irrigations a day, F2: one irrigation a day, F3: one irrigation every two days, F4: one irrigation every three days).

### 3.4. Tomato Production and Quality

3.4.1. Plant Growth

Table 3 presents the results from the ANOVA and Duncan's range multiple test of the first experiment.

The effect of water dose in plant height was significant at the contrary than the effect of irrigation frequency in the first experiment. The water dose applied by local farmers (140% ET$_c$–516 mm) and the 120% ET$_c$ (428 mm) resulted in plants with the highest heights: 179.90 and 176.32 cm, respectively. The smallest height corresponded to 80% ET$_c$ (294 mm). In the second experiment, the effect of water dose and irrigation frequency was not significant.

**Table 3.** Analysis of variance and Duncan's multiple range test of the average plant height, stem diameter, fruits per plant, total yield, marketable yield, total water use efficiency and marketable water use efficiency for the first experiment.

| Factor | Plant Height (cm) | Stem Diameter (mm) | Fruits per Plant (-) | Total Yield (kg Plant$^{-1}$) | Marketable Yield (kg Plant$^{-1}$) | Total WUE (kg m$^{-3}$) | Marketable WUE (kg m$^{-3}$) |
|---|---|---|---|---|---|---|---|
| Irrigation frequency | | | | | | | |
| F1 | 176.04 a | 11.66 a | 47.63 a | 11.01 a | 10.08 a | 49.83 a | 45.50 a |
| F2 | 175.33 a | 12.63 a | 47.50 a | 11.91 a | 11.08 a | 54.39 a | 50.48 a |
| Water doses | | | | | | | |
| L1 | 172.45 c | 10.41 c | 46.38 a | 9.62 d | 8.69 d | 58.32 a | 52.68 a |
| L2 | 174.06 bc | 11.62 b | 47.63 a | 11.00 c | 10.12 c | 54.26 ab | 49.92 ab |
| L3 | 176.33 ab | 13.18 a | 47.50 a | 11.95 b | 11.15 b | 49.89 bc | 46.56 bc |
| L4 | 179.90 a | 13.35 a | 48.75 a | 13.27 a | 12.36 a | 45.97 c | 42.81 c |
| ANOVA | | | | | | | |
| F | ns | ns | ns | ns | ns | ns | ns |
| L | ** | *** | ns | *** | *** | *** | * |
| F × L | ns | ns | ns | ns | ns | ns | ns |

F and L represent irrigation frequency and water height, respectively. F1: one irrigation per day; F2: two irrigations per day; L1: 80% ET$_c$; L2: 100% ET$_c$; L3: 120% ET$_c$; L4: 140% ET$_c$ (local farmer's dose); WUE: water use efficiency; *: significant at $p < 0.05$; **: significant at $p < 0.01$; ***: significant at $p < 0.001$; ns: no significant at $p < 0.05$. Values within the same columns that are accompanied by different letters vary significantly at $p < 0.05$.

The effect of water dose in stem diameter was significant but the effect of irrigation frequency was not significant in the first experiment. The water doses 140%ET$_c$ (13.35 mm) and 120% ET$_c$ (13.18 mm) resulted in the highest stem diameters whereas the 80% ET$_c$ produced the smallest diameter (10.41 mm). In the second experiment, the effect in both water dose and irrigation frequency was significant. Thus, the 120% ET$_c$ resulted in the highest stem diameter, and it coincides with the first experiment. Likewise, one and two irrigations per day presented the highest stem diameters whereas one irrigation every three days showed the lowest. Similar results were observed in the first experiment.

### 3.4.2. Number of Fruits per Plant

As presented in Tables 3 and 4, the effect of water dose and irrigation frequency on the number of fruits per plant was not significant in both experiments.

**Table 4.** Analysis of variance and Duncan's multiple range test of the average plant height, stem diameter, fruits per plant, total yield, marketable yield, total water use efficiency and marketable water use efficiency for the second experiment.

| Factor | Plant Height (60 DAT) (cm) | Plant Height (90 DAT) (cm) | Stem Diameter (mm) | Fruits per Plant (-) | Total Yield (kg Plant$^{-1}$) | Marketable Yield (kg Plant$^{-1}$) | Total WUE (kg m$^{-3}$) | Marketable WUE (kg m$^{-3}$) |
|---|---|---|---|---|---|---|---|---|
| Water doses | | | | | | | | |
| L1 | 118.19 a | 180.40 a | 9.93 b | 85.44 a | 6.99 b | 5.36 b | 34.59 a | 26.51 a |
| L2 | 118.82 a | 182.29 a | 11.44 a | 87.31 a | 8.25 a | 6.68 a | 34.46 a | 27.92 a |
| Irrigation frequency | | | | | | | | |
| F1 | 117.83 a | 184.55 a | 11.56 a | 87.00 a | 8.32 a | 6.77 a | 37.69 a | 30.55 a |
| F2 | 119.00 a | 183.05 a | 11.16 ab | 85.75 a | 7.68 ab | 6.13 ab | 34.73 ab | 27.71 ab |
| F3 | 117.95 a | 178.74 a | 10.37 bc | 87.25 a | 7.08 b | 5.34 b | 32.22 b | 24.26 b |
| F4 | 119.23 a | 179.04 a | 9.63 c | 85.50 a | 7.39 b | 5.84 ab | 33.45 b | 26.34 ab |
| ANOVA | | | | | | | | |
| L | ns | ns | *** | ns | *** | *** | ns | ns |
| F | ns | ns | ** | ns | * | * | * | * |
| L × F | ns | ns | ns | ns | ns | ns | ns | ns |

L and F represent water height and irrigation frequency, respectively. L1: 100% ET$_c$; L2: 120% ET$_c$; F1: two irrigations per day; F2: one irrigation per day; F3: one irrigation every two days; F4: one irrigation every three days; DAT: days after transplant; WUE: water use efficiency; *: significant at $p < 0.05$; **: significant at $p < 0.01$; ***: significant at $p < 0.001$; ns: no significant at $p < 0.05$. Values within the same columns that are accompanied by different letters vary significantly at $p < 0.05$.

### 3.4.3. Yield

In the first experiment, no significant differences in irrigation frequencies were observed, but they were observed in the second experiment. Both total and marketable yields increased as irrigation frequency increases. In both experiments, the highest water doses resulted in the highest yields (Tables 3 and 4).

In the first experiment, irrigation frequency did not affect yield. In the second experiment, the frequencies of one and two irrigations per day increased total yield and together with one irrigation every three days resulted in the highest marketable yield.

### 3.5. Tomato Quality

The effect of water dose and frequency on pH was not significant in both experiments (Tables 5 and 6).

**Table 5.** Analysis of variance and Duncan's multiple range test of the average pH, total soluble solids and titratable acidity for the first experiment.

| Factors | pH (-) | Total Soluble Solids (° Brix) | Titratable Acidity (%) |
|---|---|---|---|
| Irrigation frequency | | | |
| F1 | 4.33 a | 4.09 a | 0.27 a |
| F2 | 4.33 a | 3.93 a | 0.25 a |
| Water doses | | | |
| L1 | 4.33 a | 4.32 a | 0.27 ab |
| L2 | 4.33 a | 4.11 b | 0.28 a |
| L3 | 4.35 a | 3.88 c | 0.25 b |
| L4 | 4.30 a | 3.71 d | 0.23 c |
| ANOVA | | | |
| N | ns | ns | ns |
| L | ns | *** | *** |
| F × L | ns | ns | ns |

F and L represent irrigation frequency and water height, respectively. F1: one irrigation per day; F2: two irrigations per day; L1: 80% $ET_c$; L2: 100% $ET_c$; L3: 120% $ET_c$; L4: 140% $ET_c$ (local farmer's dose); pH: hydrogen potential; ***: significant at $p < 0.001$; ns: no significant at $p < 0.05$. Values within the same columns that are accompanied by different letters vary significantly at $p < 0.05$.

**Table 6.** Analysis of variance and Duncan's multiple range test of the average pH, total soluble solids and titratable acidity for the second experiment.

| Factor | pH (-) | Total Soluble Solids (° Brix) | Titratable Acidity (%) |
|---|---|---|---|
| Water doses | | | |
| L1 | 4.20 a | 5.36 a | 0.28 a |
| L2 | 4.22 a | 5.12 b | 0.26 b |
| Irrigation frequency | | | |
| F1 | 4.25 a | 5.22 a | 0.28 a |
| F2 | 4.22 a | 5.24 a | 0.28 a |
| F3 | 4.19 a | 5.30 a | 0.28 a |
| F4 | 4.20 a | 5.20 a | 0.26 a |
| ANOVA | | | |
| L | ns | * | * |
| F | ns | ns | ns |
| L × F | ns | ns | ns |

L and F represent water height and irrigation frequency, respectively. L1: 100% $ET_c$; L2: 120% $ET_c$; F1: two irrigations per day; F2: one irrigation per day; F3: one irrigation every two days; F4: one irrigation every three days; DAT: days after transplant; WUE: water use efficiency; *: significant at $p < 0.05$; ns: no significant at $p < 0.05$. Values within the same columns that are accompanied by different letters vary significantly at $p < 0.05$.

In the first experiment, TSS were affected by irrigation frequency, but they were not affected by water doses (see Table 5). The highest Brix degrees was observed in 80% $ET_c$

and the smallest in 140% $ET_c$. In the second experiment, irrigation frequencies did not affect TSS either. The $ET_c$ 100% showed higher values than 120% $ET_c$.

Likewise, the effect of water dose on titratable acidity was significant in both experiments although the effect of irrigation frequency was not significant (see Tables 5 and 6). Three ranges of citric acid percentage were observed. In the first experiment, the 100% ETc and 80% $ET_c$ doses resulted in the highest percentages: 0.28% and 0.27%, respectively, and the lowest (0.22%) was observed in 140% $ET_c$. In the second experiment, the 100% $ET_c$ resulted in higher citric acid percentage than 120% $ET_c$.

### 3.6. Water Use Efficiency

In the first experiment, no significant differences in irrigation frequencies were observed, but water doses significantly differed. In the second experiment, no significant differences were detected in water doses, but irrigation frequencies differed . As shown in Table 3, the 80% $ET_c$ and the 100% ETc doses resulted in the highest WUE and the 140% $ET_c$, the lowest in the first experiment. As shown in Table 4, the two and one irrigation per day achieved the highest total WUE in the second experiment. Likewise, the highest marketable WUE was achieved in the frequencies: one and two irrigations per day, and one irrigation every three days.

## 4. Discussion

### 4.1. Soil Moisture on Tomato Growth and Yield

Plants need water to fulfil their physiological stages [51]. Soil moisture is a key factor for photosynthesis, crop productivity and water use efficiency [52]. Figures 3 and 4 show that as higher the water dose as higher soil moisture. Likewise, soil moisture variation was higher in the large irrigation intervals. Therefore, plant growth will be affected, the energy applied for plant's roots to absorb water and dissolved nutrients increases as soil water decreases. Proper irrigation scheduling will balance water dose and irrigation frequency in order to maintain soil moisture within a proper threshold for crop production and quality [37].

### 4.2. Water Doses and Irrigation Frequency on Crop Production, Crop Quality and WUE

Plant height, stem diameter and tomato yield decreased as water dose decreases. These results agree with [21,29], who observed a decrease in vegetative growth and fruit yield in deficit irrigations. Likewise, Ref. [53] reported a negative effect of water stress on plant height and stem diameter.

Neither water doses nor irrigation frequencies affected the number of fruits per plant. These results coincide with [54], who observed the same number of fruits per plant in 100% $ET_c$ and 75% $ET_c$. Conversely, Ref. [21,55] concluded that water dose affects fruit number. Flower abortion may result on lower number of fruits per plant; tomato is highly sensitive to water stress, especially during flowering and fruiting [56]. Thus, water stress could reduce fruit number and fruit weight [26]. However, in the present study no decrease in fruit due to water stress was observed.

Water doses significantly affected tomato yield and these findings agree with previous studies which highlighted that small water doses negatively affect fruit yield [21,29,37,57]. Moreover, Ref. [58] observed the highest tomato yield in the highest water dose (150% $ET_c$), which coincides with our results.

The effect of irrigation frequency on yield has been statistically significant and agrees with [36], who observed the highest yields in the highest irrigation frequencies (2 days) in tomato crops. Nevertheless, Ref. [40] evaluated five irrigation frequencies (1, 3, 5, 7 and 9 days) in tomatoes grown in open fields and observed that the weekly frequency resulted in the maximum yield while the daily irrigation resulted in the minimum.

Likewise, Ref. [40] pointed out that frequent irrigations resulted in higher nutrient leaching from the root's zone. In addition, the root system develops less as irrigation frequency increases. They also argued that its development reinforces in large irrigation

intervals thus, secondary root branching and main root deepening will improve, as well as water and nutrients uptake.

The present study showed a slight difference between the total and marketable yields. In the last, one irrigation every three days and one and two irrigations per day yielded the same. However, in the total yield, one irrigation a day and every two days resulted in the highest values.

Regarding fruit quality, pH was not significantly affected by water dose and irrigation frequency. These results do not agree with those obtained by [1], who observed a decrease in pH as water doses decreases. Likewise, TSS and titratable acidity increased as water doses decreases. These results agree with previous studies and shows that tomato quality can improve under deficit irrigation [13]. The soluble solids content and soluble sugar increase as water doses decreases [21]. Similar results were obtained by [1,24,26,27,54,59–62]. These authors obtained the highest total soluble solids in the lowest water dose. Likewise, titratable acidity increased as water dose decreased [1,24,27,60]. Under soil water deficit conditions, water flow from the xylem towards the fruit could reduce [63,64]. Thus, the translocation of phloem sap to fruit is impeded and solute concentration in the sap increases and, fruit quality improves as well [63,65]. On the contrary, water accumulation in the fruit causes the dilution of fruit elements [14].

Similarly, irrigation frequency did not affect quality parameters in agreement with [66]. This author did not observe a significant effect of two irrigation frequencies (irrigation every 2 and 3 days) in soluble solids either. As irrigation frequency reduces, water stress ameliorates and the production and transfer of photosynthetic products (as a sucrose to reproductive organs) improve. Hence, fruit sugar soluble content increases although its effect is not always significant [37].

The best WUE was observed in the lowest irrigation dose (first experiment) and the results agree with [14,21,60,67–70]. However, these results do not coincide with the ones from [22,24,37,71]. These contradictory results may be explained since any irrigation (excess/deficit) tends to decrease both yield and WUE [26].

The highest WUE was obtained in the highest irrigation frequency and coincides with [34] who argues that as irrigation frequency increases, the number of fruits increases, therefore yield increases too and WUE improves. However, in the present study the number of fruits per plant was not affected neither by irrigation frequency nor by water dose. In our case, the highest yields and WUE were observed in two irrigations a day with more than 10% of big-sized fruits than the other frequencies. On the contrary, the lowest WUE was observed in an irrigation every other day since it encompasses more unmarketable fruits, between 10% and 21%, than the others.

## 5. Application to Local Farmers

This study has shown that the local growers' dose (140 $ET_c$) is not recommended to balance tomato quantity and quality and WUE. The local dose produces higher yield but lesser fruit quality and WUE than the other doses study. Likewise, the effect of irrigation frequency is small compared to water doses.

Among the water doses, the 100% $ET_c$ reaches a higher yield and the same WUE than the 80% $ET_c$. In addition, it will reduce the local farmer's dose in 40% and it will achieve a higher content of Brix degrees and the same percentage of citric acid than the 80% $ET_c$. The application of the 120% $ET_c$ dose will result in higher yield and the same water use efficiency than the 100% $ET_c$ and it will also reduce 20% of water with respect to the local farmer's dose. Likewise, the 100% $ET_c$ will have higher content of Brix degrees and citric acid than the 120% $ET_c$ and the local farmer's dose.

If farmers' interest, consumer's health and environment sustainability are taking into account, it would be advisable to apply 100% $ET_c$ in greenhouse tomato. This dose showed a balance between production, nutritional quality and water use efficiency. However, if tomato growers' income is a priority, the 120% $ET_c$ dose will be advisable, since it will provide a proper WUE and will used less water than the local farmer's dose. Furthermore,

regarding marketable yield, in both experiments the 120% ETc dose yielded, on average, 1.2 kg plant$^{-1}$ more than 100% ET$_c$ dose thus, since the greenhouse contains 17,857 plants ha$^{-1}$, it resulted in 21,428 kg ha$^{-1}$, this minus 5% field losses leaves a total of 20,357 kg ha$^{-1}$. Likewise, in the wholesale market of Ibarra (province of Imbabura), the average price per kg of tomato is 0.60 USD [72]; so, the water increase in 20% will result in an increase in farmer's income by 12,214 USD per ha, which will be very beneficial to help them to accomplished further agricultural activities.

For practical purposes, it would be advisable to apply the 100% ET$_c$ and 120% ET$_c$ doses with a frequency of one irrigation per day.

### 6. Conclusions

Tomato evapotranspiration was highly variable since climatic conditions were variable too. Hence, it would be advisable to make daily irrigation schedules to better adjust water doses.

Tomato yield, fruit quality and water use efficiency are more affected by water doses than irrigation frequency. Water doses affected all the variables studied, with the exception of fruits per plant and pH. However, irrigation frequency did not affect any of the quality variables and only its effect was significant on stem diameter, yield and water use efficiency.

Both total and marketable yields showed a proportional relationship with water doses. The highest yield increment was observed between 80% ET$_c$ and 100 ET$_c$ doses.

A daily water dose, which fulfills 100% crop evapotranspiration, is recommended to local tomato growers. This recommendation takes into account the balance among crop production, fruit quality and water use efficiency. Therefore, the grower's income will not be affected.

Water doses affect soil moisture content. Frequent irrigation minimizes temporal fluctuations in soil water content and minimizes the risk of tomato water stress. Thus, an adequate water dose applied in short irrigation intervals will benefit crop production and crop quality.

**Author Contributions:** J.E.C.-L.: conceptualization, methodology, investigation, formal analysis, writing—original draft preparation. L.R.-S.: conceptualization, supervision, writing—reviewing and editing. S.Z.-M.: supervision. All authors have read and agreed to the published version of the manuscript.

**Funding:** This research received no external funding.

**Institutional Review Board Statement:** Not applicable.

**Informed Consent Statement:** Not applicable.

**Data Availability Statement:** The data presented in this study are available on request from the corresponding author. The data are not publicly available due to their relevance to an ongoing Ph.D. thesis.

**Acknowledgments:** Javier Ezcequiel Colimba-Limaico wants to thank Secretaría de Educación Superior, Ciencia, Tecnología e Innovación (SENESCYT) and Instituto de Fomento al Talento Humano, of the Government of the Republic of Ecuador for supporting its PhD studies.

**Conflicts of Interest:** The authors declare no conflict of interest.

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
