# Peer review of "Optimal Irrigation Scheduling for Greenhouse Tomato Crop (Solanum Lycopersicum L.) in Ecuador"

_agronomy, doi:10.3390/agronomy12051020_

Round 1

Reviewer 1 Report

In my opinion, the article Optimal irrigation scheduling for greenhouse tomato crop (Solanum lycopersicum L.) in Ecuador is written correctly. The authors have clearly specified the aim of the study. Their discussion of the results is sufficient. Moreover, they also related the experimental results to the region's current agricultural practices, which is a great advantage. Therefore, in my opinion, the article is suitable for publication after taking into account minor comments:

In the introduction, please expand the description regarding tomato cultivation in Ecuador. In addition, please provide more information on the size of the cultivation, the climatic conditions, and the problems related to the production of the plant in this country.

In the methodology, please provide the exact Kc values used for the calculations and their source. Without giving the coefficients, it is not possible to check the correctness of your calculations.

Please replace references that are not in English with others. References to literature in a language other than English are of little use to most readers. Therefore, please try to include only necessary items in a language other than English.

On line 130, please use superscript when writing m2.

Reviewer 2 Report

General comments: The experiment deals with the recommendation of adequate management for irrigation in tomatoes grown in a greenhouse, depending on the depth and frequency of application. Qualitative and quantitative parameters were evaluated to support the conclusions. The manuscript is well written, clear, and has relevant results within the scope of the journal. All listed references are cited in the text, however, only 25% are from the last 5 years. Self-citations were not identified.

Specific comments: Please, see attached the PDF version of the manuscript with some suggestions and comments. 
